# Inhibition of *Candida albicans* and Mixed Salivary Bacterial Biofilms on Antimicrobial Loaded Phosphated Poly(methyl methacrylate)

**DOI:** 10.3390/antibiotics10040427

**Published:** 2021-04-13

**Authors:** Andrew R. Dentino, DongHwa Lee, Kelley Dentino, Arndt Guentsch, Mohammadreza Tahriri

**Affiliations:** 1Department of Surgical Sciences, Marquette University School of Dentistry, Milwaukee, WI 53201-1881, USA; evergreenx@yahoo.com (D.L.); kelley.dentino@marquette.edu (K.D.); arndt.guentsch@marquette.edu (A.G.); 2Department of Engineering, Norfolk State University, Norfolk, VA 23504, USA; mtahriri@nsu.edu

**Keywords:** *Candida albicans* biofilm, mixed salivary bacterial biofilm, phosphated PMMA, antimicrobials, salivary pellicle

## Abstract

Biofilms play a crucial role in the development of *Candida*-associated denture stomatitis. Inhibition of microbial adhesion to poly(methyl methacrylate) (PMMA) and phosphate containing PMMA has been examined in this work. *C. albicans* and mixed salivary microbial biofilms were compared on naked and salivary pre-conditioned PMMA surfaces in the presence or absence of antimicrobials (Cetylpyridinium chloride [CPC], KSL-W, Histatin 5 [His 5]). Polymers with varying amounts of phosphate (0–25%) were tested using four *C. albicans* oral isolates as well as mixed salivary bacteria and 24 h biofilms were assessed for metabolic activity and confirmed using Live/Dead staining and confocal microscopy. Biofilm metabolism was reduced as phosphate density increased (15%: *p* = 0.004; 25%: *p* = 0.001). Loading of CPC on 15% phosphated disks showed a substantial decrease (*p* = 0.001) in biofilm metabolism in the presence or absence of a salivary pellicle. Salivary pellicle on uncharged PMMA enhanced the antimicrobial activity of CPC only. CPC also demonstrated remarkable antimicrobial activity on mixed salivary bacterial biofilms under different conditions displaying the potent efficacy of CPC (350 µg/mL) when combined with an artificial protein pellicle (Biotene half strength).

## 1. Introduction

The formation of biofilms on implants and devices used in dentistry and medicine has been known to cause significant morbidity [1,2,3]. On poly(methyl methacrylate) (PMMA) materials in particular, the adhesion of *Candida albicans* has been recognized as a problem for decades [4,5,6], and along with poor denture hygiene is thought to play a central role in the development of *Candida*-associated denture stomatitis (CADS) [7,8,9,10,11,12].

Prevalence rates of CADS vary widely (18–65%) depending on the population studied, with higher rates often seen in the institutionalized elderly [4,13]. Current treatments for CADS include the use of topical antifungal agents repeatedly applied to both the mucosa and the denture surface, but the re-infection rate is extremely high. Regular professional oral care, including simply removing dentures at night and leaving them to dry in the air, results in a decrease in the number of positive *Candida* cultures as well as a decrease in oral mucosal inflammation [14]. Moreover, incorporating professional oral care, including denture hygiene, resulted in a lower incidence of pneumonia, and a decrease in the number of febrile days in this frail population [15].

Such observations have spurred an increase in research on material alterations in an attempt to reduce *Candida* adhesion and subsequent biofilm formation on PMMA and non-PMMA based surfaces [16,17,18,19,20,21,22].

Poly(methyl methacrylate) (PMMA) is the most commonly used material for the fabrication of dentures. However, it has been speculated that PMMA has a limitation in terms of its surface properties, in particular, the absence of ionic charge which may be required for selective adsorption of salivary antimicrobials such as defensins and histatins [23,24]. Since salivary antimicrobials are cationic peptides, they are strongly adsorbed onto the tooth surface by electrostatic interaction. On the other hand, absence of negative charge on the PMMA surface not only minimizes the adsorption of defense molecules, but the attractive London-van der Waals forces actually facilitates the adherence of *C. albicans* on the denture surface, leading to denture stomatitis [25,26].

It has been demonstrated that carboxylated PMMA derivatives provide a negative charge and inhibit *C. albicans* adhesion in vitro and enhance adsorption of cationic salivary antimicrobial peptides onto the surface [16,17,27].

We postulated that addition of phosphate would likewise provide the required surface properties to PMMA polymers that would yield a phosphate density-dependent decrease in *C. albicans* adherence to the negatively charged denture base material compared to the normal uncharged PMMA. We further examined the surface charge effects on the adhesion and biofilm formation of fresh mixed salivary bacteria, as well as the effects of several cationic antimicrobials exposed to the various PMMA polymers in the presence or absence of natural or artificial salivary pellicles.

The goals of this in vitro study were three-fold. First, to assess the effects of phosphate charge to inhibit microbial colonization. Second, to evaluate the effects of cationic antimicrobial loading to further inhibit microbial adhesion, and third, to determine the effects of protein pellicle to alter microbial adhesion and biofilm development in the presence or absence of cationic antimicrobials on charged and uncharged PMMA surfaces.

We have employed a model system using PMMA disks processed with varying levels of phosphate monomers and polished to an R_a_ level simulating the in taglio surface of a denture. Biofilm formation by four different clinical *C. albicans* isolates (referred to as A1, A3, AD1 and 1.1) and fresh pooled salivary bacteria was assessed visually using confocal microscopy with live/dead staining as well as quantitatively using the reduction of a tetrazolium salt (XTT) to measure cellular metabolic activity colorimetrically. The parameters of anionic charge, cationic antimicrobial loading and protein pellicle effects were evaluated for their ability to alter microbial adhesion and early biofilm development on denture base materials in this 24-h biofilm model system.

## 2. Results

**Effects of Anionic Charge:** The mean biofilm respiration/XTT values of the *Candida* biofilms (AD1) on 0, 5, 15, and 25% phosphated PMMA surfaces were 0.195 ± 0.009, 0.189 ± 0.008, 0.176 ± 0.004, and 0.173 ± 0.007, respectively. The one-way ANOVA indicated that there was a phosphate dependent decrease in *Candida* biofilm metabolic activity with 15 and 25% phosphated PMMA groups reaching statistical significance (15%: *p* = 0.004; 25% *p* = 0.001) (Figure 1A). The confocal images mirrored the XTT results showing a consistent but modest drop in biofilm development with the addition of phosphate charge. There was however an apparent plateau in biofilm reduction from 15–25% Phosphate (Figure 1B, Appendix A). There was less inhibition by charge in the mixed salivary bacteria (MSB) model as seen in Figure 2A,B where the naked PMMA and 15% Phosphate PMMA unloaded surfaces showed no significant drop in biofilm respiration in the MSB experiments as compared to the significant, albeit modest drop in the *C. albicans* 1.1 strain. From this point we limited our experiments to the 15% phosphate disks in comparison to the control and began to evaluate the effects of antimicrobial loading of the surfaces.

**Effects of Antimicrobial Loading:**Figure 2A demonstrates that KSL-W and Histatin 5 preloaded on either PMMA or 15% phosphated PMMA at 100 µg/mL showed modest, but significant inhibition against *C. albicans*, but no significant reduction in the MSB model where there was more variability. In contrast, loading 100 µg/mL CPC on 15% phosphated PMMA disks resulted in a very significant decrease in *Candida* biofilm as well as MSB biofilm metabolic activity (Figure 2A, CPC loaded). The striking and consistent finding for both fungal and bacterial inocula led us to focus our efforts on further describing the CPC activity rather than fine tune our observations on KSL-W and Histatin 5.

**Charge and Natural Pellicle Effects:** Unlike KSL-W and Histatin 5 loaded disks where surface charge effects at this concentration were unremarkable (Figure 2A), CPC showed a significant antimicrobial effect on *Candida* as well as mixed salivary bacterial biofilm metabolism when loaded onto the naked charged surface, whereas CPC efficacy was consistently modest when loaded onto a naked uncharged PMMA surface (Figure 2A,B). However, in the presence of a salivary pellicle, the antimicrobial efficacy of CPC at 100 µg/mL was enhanced in a manner that created an antimicrobial surface that was indistinguishable from the CPC-loaded 15% phosphated PMMA (Figure 2B and Figure 3F,G). This was consistent for all *Candida* and MSB inocula. Results for all four *Candida* strains are shown in Appendix A.

Figure 3 confirms that the 15% phosphate PMMA reduces *Candida* strain AD1 adhesion (Figure 3A,B), and also demonstrates that a natural salivary pellicle reduces if not completely abrogates the surface charge inhibition of *Candida* adhesion and biofilm development in this model system (Figure 3B,D). In addition, it provides striking visual confirmation of a microbial free surface (Figure 3F–H) in the presence of CPC consistent with the Figure 2B biofilm metabolism assays from strain 1.1 (CPC + Saliva condition).

**Artificial Pellicle and CPC Enhancement Effects:** CPC also showed significant antimicrobial activity on *Candida* adhesion and biofilm formation when it was combined with an artificial protein pellicle (Biotene PBF; Glaxo Smith Kline) (Figure 4A,B).

CPC (350 µg/mL) also showed consistent and significant antimicrobial activity on mixed salivary bacterial biofilms when combined with an artificial protein pellicle (Biotene PBF half strength) (Figure 5A,B). The Biotene and CPC combination was indistinguishable from the CPC loaded 15% phosphated PMMA and the negative controls of heat killed MSB and no inoculum.

## 3. Discussion

Creating surfaces that resist microbial fouling is of great interest, particularly in dentistry where tooth loss often results in removable prosthetics which are known to contribute to morbidity in the institutionalized elderly [12,15]. PMMA is a favored material for denture base production because of its attractive properties [28,29,30]. Be that as it may, this material is easily and repeatedly colonized by different microbial species, including *C. albicans*, *C. glabrata* and gram positive/negative organisms, particularly on the unpolished in taglio surface. Various endeavors have been made to overcome this drawback of denture base resins [31]. There have been efforts ranging from the use of over-the-counter cleansing agents and surfactants [32,33], to employing electrostatic interactions to reduce adhesion [5,12,17,25,26,27], along with combinations of charge alteration and antimicrobial loading [16,34]. More recently nanotechnology is being employed with titanium dioxide [29] or silver [35] incorporated within PMMA polymers as well as the development of CPC in a colloidal nanocarrier of Chitosan and Iron Oxide [36].

In this study, we examined phosphate-containing PMMA polymers for their ability to inhibit the adherence of *Candida albicans* and mixed salivary bacteria based on phosphate charge density. We subsequently characterized the effects of antimicrobial loading and natural and artificial saliva pellicle effects on microbial adhesion in a 24 h in vitro biofilm model.

Regarding charge effects, we have been able to demonstrate that phosphated-PMMA can reduce microbial adhesion similar to carboxylated PMMA [17,27,37], but the effect in this model system is limited and is more pronounced for *C. albicans* strains than for the mixed salivary bacteria. Both the 15% and 25% disks showed a reduction in *Candida* biofilm development in comparison to the control and the 5% Phosphate samples (Figure 1 and Appendix A).

As observed in carboxylated PMMA [34] we also noted for phosphated PMMA the charge density inhibition of microbial adhesion became less dramatic overtime if the disks were not cleaned scrupulously after each experiment. This corresponded to an increase in surface roughness as assessed by profilometry, and it was most pronounced in our 25% phosphate disks. We solved this fouling problem by changing our disk cleaning procedures and were able to regenerate charged surfaces and eliminate issues with reproducibility or longevity of the disks with that methodological change. These findings do, however, dampen the commercial prospects for such technology since compliance with a rigorous regeneration protocol seems unlikely.

After demonstrating a significant, but not overwhelming charge dependent reduction in biofilm metabolism, we began investigating the inhibitory effects of antimicrobials loaded onto the 15% phosphated PMMA and uncharged PMMA disks.

We limited our phosphate levels to 15% for the subsequent experiments because water sorption data [38] suggested that this technology may have trouble meeting the water sorption ISO standard at higher levels of phosphate.

Based on previous work with Histatin 5 [34] and KSL-W [39], we anticipated that phosphated PMMA polymers would adsorb these peptide antimicrobials onto the surface and show a strong inhibitory effect on *Candida* adherence and biofilm development.

The antimicrobials we focused on were Histatin 5, KSL-W and CPC because of their antimicrobial efficacy against oral microbes and their cationic nature. Employing antimicrobial peptide concentrations previously shown to be effective on oral bacteria and *C. albicans* respectively [34,40,41], we included the widely used antiseptic CPC [42] at the same concentration and have demonstrated that CPC had the most pronounced anti-adhesion effects on a charged surface compared to KSL-W and Histatin 5 under the conditions of our assay (Figure 2A and Figure 3). The observation that Histatin 5 showed significant reductions on uncharged PMMA is consistent with other studies looking at *C albicans* survival on polystyrene and PMMA with a Histatin 5 pellicle [34,43] as well as some smaller peptide variants of Histatin 5 on PMMA [41]. Differences in the assay conditions make comparisons difficult, however these studies provided the rationale for our use of 100 μg/mL concentrations for the initial loading experiments. It should be noted that levels of Histatin 5 in whole saliva range from 2–8 μg/mL [44].

The anti-adhesion effects of natural salivary pellicle combined with CPC on PMMA were somewhat unexpected. Figure 2B and Figure 3 provided strong evidence that saliva enhanced the efficacy of CPC in both charged and uncharged PMMA. Saliva seemed to retain and stabilize the efficacy of CPC on PMMA, and pellicle shielding of the net negative surface charge on the 15% phosphate PMMA turned out to be inconsequential. A strong rechargeable antimicrobial shield was created on both surfaces. The effect was consistent across our *C. albicans* isolates as well as the MSB model (Figure 2B and Figure 3).

This finding seemed likely to have clinical relevance for denture hygiene, but with many of the institutionalized elderly facing the challenge of decrease in salivary flow resulting in xerostomia (dry mouth), we decided to evaluate the potential for replicating the antimicrobial shield on PMMA surfaces, charged or uncharged, using an artificial saliva to generate a pellicle. We chose Biotene PBF because it was a popular over-the-counter salivary substitute. Figure 4 and Appendix A demonstrate consistency in the ability of the Biotene PBF formulation to create such an antimicrobial pellicle shield when using CPC at concentrations that are found in over the counter mouthrinses. These experiments also demonstrated that *Candida* strains showed differential responses to Biotene PBF pellicle or CPC loaded (350 µg/mL) disks on the uncharged PMMA surface. Figure 5A,B confirmed that the combined protein/CPC based pellicle was consistently effective in the mixed salivary bacteria model as well.

Based on previous work with Histatin 5 [34] and KSL-W [39], we anticipated that phosphated PMMA polymers would more strongly adsorb these peptide antimicrobials onto the surface and show an increased inhibitory effect on *Candida* adherence and biofilm development. However, the antimicrobial effects of the cationic peptides were marginal in this system (Figure 2A). In contrast, CPC showed statistically significant, and potentially clinically relevant reductions in the adhesion/biofilm metabolism of *C. albicans* on 15% phosphated PMMA. Although Histatin 5 and KSL-W did show statistically significant reductions compared to unloaded PMMA controls, the level of inhibition was unimpressive compared to CPC on a charged surface where the metabolic activity and confocal microscopic images showed essentially no biofilm formation. These conditions were indistinguishable from heat-killed and non-inoculated controls.

Biofilms on restorative materials are common, problematic, and fairly well characterized on multiple different materials including PMMA as recently reviewed by Schmalz and Cieplik [45]. CPC (Cetylpyridinium chloride) is a cationic quaternary ammonium compound with an extended aliphatic chain. It can interact with the bacterial cell membrane, resulting in a leakage of cellular components, disruption of cellular metabolism, inhibition of cell growth and cell death. However, it can also promote microbial resistance which is something that must also be evaluated over time [42]. The data presented here suggest that it may be worthwhile to develop clinical trials to assess the regular use of a protein/CPC pellicle after physically cleaning dentures as part of regular denture hygiene with the aim of reducing aspiration pneumonia in the denture wearing institutionalized elderly.

## 4. Experimental Procedure

### 4.1. Polymer Synthesis and Fabrication of Phosphated PMMA Disks

Four polymers with varying amounts of phosphate (0%, 5%, 15%, 25%) were synthesized by monomer substitution using Lucitone 199 denture base polymer beads and mixtures of methyl methacrylate (MMA) and ethylene glycol methacrylate phosphate (EGMP) monomers (Sigma-Aldrich, Milwaukee WI, USA). The ratios of MMA/EGMP were 100:0 (PMMA control), 95:5 (5% EGMP), 85:15 (15% EGMP), and 75:25 (25% EGMP), respectively. Disks (diameter: 15 mm, thickness: 3 mm) were heat-processed according to the manufacturer’s instructions and polished to 2 mm thickness with 600 grit silicon carbide metallographic grinding paper (PACE Techonologies, Elkridge, MD, USA). The mean surface roughness (Ra) of the disks was 0.40 µm. Disks were stored in sterile distilled water until used. Regeneration of disks involved a 10-min sonication in tap water, then repeated washes in deionized water (5X), a 1 h RBS detergent soak, a repeat of the deionized water wash (5X), and finally an overnight soak in 0.05% trypsin w/0.053 mM EDTA. Disks were then sonicated 10 min in ETOH followed by repeated deionized water washes (5X).

### 4.2. Candida and Mixed Salivary Bacterial Adhesion and Biofilm Metabolism on Phosphated PMMA Surfaces

Four clinical isolates of *C. albicans* (A1, A3, AD1, 1.1) were obtained from various sources and identified through germ tube formation. Strain A1 was a sputum sample. Strain A3 was obtained from a bronchial lavage sample. Strains AD1 and 1.1 were oral isolates from denture stomatitis patients. *C. albicans* cells were grown in YPD growth medium (yeast extract 10 g, peptone 20 g, dextrose 20 g in 1000 mL water) at 37 °C overnight. Cells were harvested, washed with phosphate-buffered saline, and standardized to 1 × 10^7^ cells/mL. Fresh mixed salivary bacteria and 50% saliva buffer were obtained as described [40]. Briefly, 3–4 healthy human donors provided unstimulated saliva the day of an experiment which was pooled, centrifuged at 500 RPM for 10 min at 4°C to remove epithelial cells. The supernatant was collected and re-spun at 4000 RPM for 15 min at 4°C. The supernatant was then filtered through a 0.2 µm Millipore filter, and that saliva preparation was mixed 1:1 with sterile PBS to serve as 50% saliva buffer for pellicle formation on disks. The bacterial pellet was washed 3 times and resuspended in PBS and standardized to 1 × 10^7^ cells/mL.

Six disks from each phosphate density group (0%, 5%, 15%, 25%) were selected as samples. The polished disk was covered with 60 µL of standardized *C. albicans* suspension and all the samples were incubated at 37 °C for 90 min to allow the *Candida* cells to adhere to the disk surfaces. Following this inoculation procedure, the disks were gently rinsed with sterile distilled water to remove non-adherent cells and placed in wells of a 12-well tissue culture plate containing SD media (yeast nitrogen base 1.7 g, ammonium sulfate 5.0 g, dextrose 20 g in 1000 mL water). The plates were incubated at 37 °C for 24 h. The disks were then transferred to the wells of a 12-well tissue culture plate containing 2 mL fresh 2,3-bis(2-methoxy-4-nitro-5-sulfo-phenyl)-5-[(phenyl amino) carbonyl]-2H-tetrazolium hydroxide (XTT). The plates were incubated for 3 h at 37 °C. The entire contents of the well were transferred into a 2 mL tube and centrifuged (5 min, 6000 g). From each tube, 200 µL XTT formazan in the supernatant was transferred to one of the wells of the 96-well microtiter plate for XTT assay [18]. Metabolic activity of *Candida* biofilm on each disk was estimated using a microtiter plate reader (PowerWave™ XS, BioTek Instruments, Inc. Burlington, VT, USA) at 492 nm. Negative controls included no inoculation as well as heat-killed *C. albicans or MSB*. *Candida* or MSB biofilms on polystyrene disks served as a positive control. Statistical significance was analyzed using a one-way ANOVA at the 95% confidence level to determine if the means of the biofilm metabolic activity were significantly different between the disks with different phosphate density.

### 4.3. Microbicidal Activity of Antimicrobials on Oral Microorganisms

The anti-adhesion effects of Histatin 5, KSL-W, and Cetylpyridinium chloride (CPC) on *C. albicans* as well as mixed salivary bacterial biofilms were assessed. Antimicrobials were obtained at >95% purity from Sigma Aldrich (Milwaukee, WI, USA). Twenty-four 0% phosphated PMMA disks and twenty-four 15% phosphated PMMA disks were pre-conditioned for 1 h at room temperature with 50% saliva buffer prepared as described [40]. Non-preconditioned disks served as control. The disks were then either loaded or not loaded with Histatin 5, KSL-W or CPC (100 µg/mL) for 60 min at room temperature. Subsequently, *C. albicans* (1 × 10^7^ cells/mL) or fresh mixed salivary bacteria (1 × 10^7^ cells/mL) was inoculated on the disks and incubated at 37 °C for 90 min. Afterward, the disks were gently rinsed with sterile distilled water to remove non-adherent cells. Subsequently, the disks were placed in 12-well tissue culture plates containing SD media (for *C. albicans*) or Todd-Hewitt Broth (for mixed salivary bacteria) and then incubated at 37 °C for 24 h. The biofilms developed on the disks were subjected to either confocal assessment after live dead staining or an XTT assay to evaluate their metabolic activity. Statistical analysis using a multifactorial ANOVA was performed on the data to determine if there was a significant difference between the different groups.

### 4.4. Image Analysis

A confocal microscope (Nikon Eclipse Ti, Nikon Instruments Inc., Melville, NY, USA) with a 40X lens (pinhole size 69.0 µm) was used to record confocal image stacks of *Candida* biofims and mixed salivary bacterial biofilms in five random locations near the center of each disk. The abundance and viability of the biofilms were analyzed with confocal microscopy using a commercially available Live/Dead stain kit (Molecular Probes ThermoScientific, Eugene, OR, USA), with green fluorescence indicating live microbes and red fluorescence indicating dead microbes.

## 5. Conclusions

New phosphate containing EGMP-PMMA co-polymers are capable of altering *C. albicans* adhesion and/or biofilm metabolism.

Pre-conditioning a negatively charged 15% phosphated PMMA surface with CPC provides a strong antimicrobial shield whereas the uncharged PMMA surface is not able to benefit from CPC exposure in this in vitro model system.

A protein pellicle coating provides the necessary environment to establish an effective antimicrobial shield on a PMMA surface while not altering the phosphated PMMA from efficacious delivery of CPC.

The potential of these findings to improve denture hygiene in the institutionalized elderly should be explored further in clinical trials.

## Figures and Tables

**Figure 1 antibiotics-10-00427-f001:**
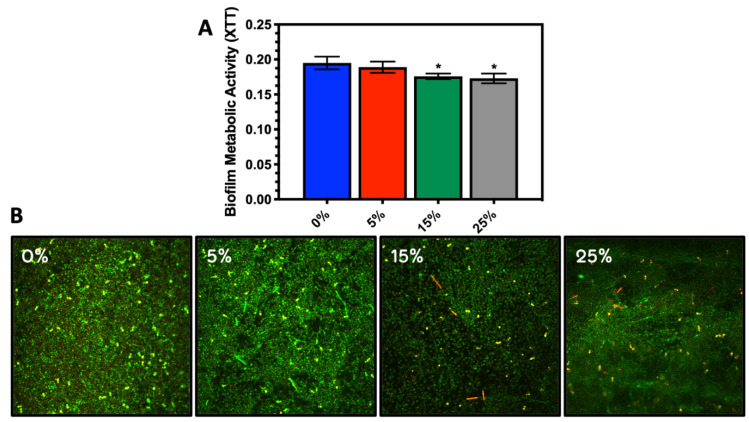
(**A**) Metabolic activity of the biofilms as measured by XTT. Average values and standard deviations were averaged from 6 replicate disks for each condition and (**B**) Confocal microscopic images of *Candida* (AD1) biofilm of on 0, 5, 15, and 25% phosphated PMMA using Live/Dead staining. * Statistically significant reduction in metabolic activity compared to PMMA control (0% Phosphate).

**Figure 2 antibiotics-10-00427-f002:**
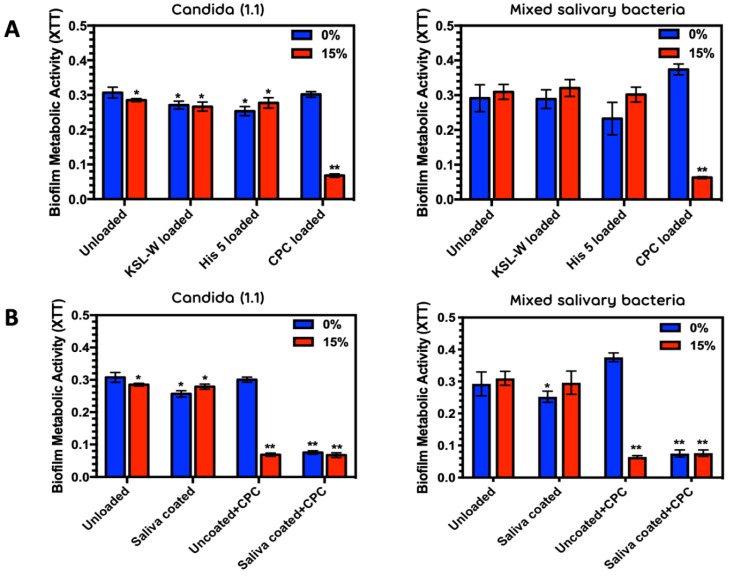
(**A**) Effects of Histatin 5, KSL-W, and CPC on *Candida* (1.1) biofilm metabolism and mixed salivary bacteria biofilm metabolism on 0 and 15% phosphated PMMA and (**B**) Antimicrobial effect of CPC on *Candida* (1.1) biofilm metabolism and mixed salivary bacterial biofilm metabolism in the presence or absence of a salivary pellicle on 0 and 15% phosphated PMMA. Average values and standard deviations were taken from six replicate disks for each condition. *;** Statistically significant reduction in metabolic activity compared to PMMA control (0% Phosphate).

**Figure 3 antibiotics-10-00427-f003:**
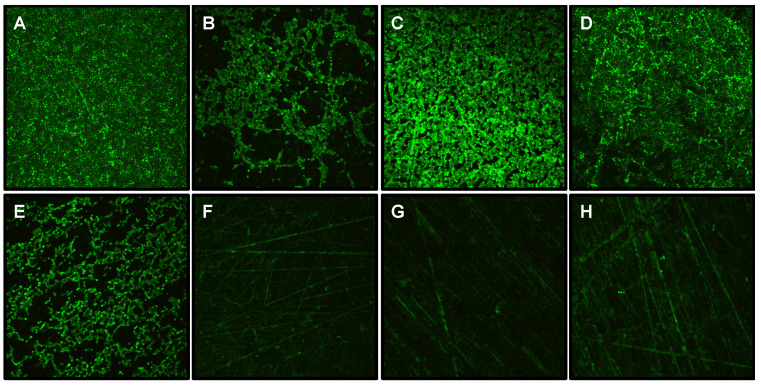
Confocal microscopic images of *Candida* (AD1) biofilms on naked or saliva-coated 0 and 15% phosphated PMMA (24 h, 40X); (**A**) 0%, naked, (**B**) 15%, naked, (**C**) 0%, saliva-coated, (**D**) 15%, saliva-coated, (**E**) 0%, CPC-loaded, (**F**) 15%, CPC-loaded, (**G**) 0%, saliva + CPC and (**H**) 15%, saliva + CPC. CPC at 100 µg/mL.

**Figure 4 antibiotics-10-00427-f004:**
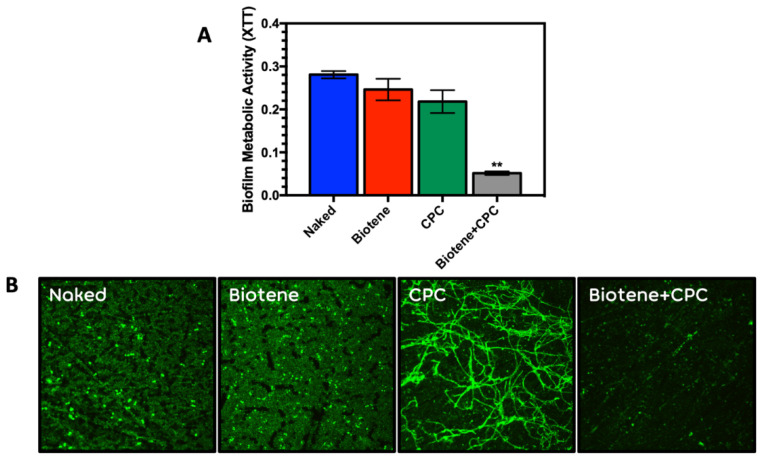
(**A**) Metabolic activity and (**B**) Confocal microscopic images of *Candida* (AD1) biofilm metabolism on the naked PMMA surface, and 3 artificial pellicles, Biotene PBF preconditioned, CPC preconditioned (350 µg/mL) and the combination of Biotene PBF + CPC. Average XTT values and standard deviations were determined from 6 replicate disks for each condition. ** Statistically significant reduction in metabolic activity compared to PMMA control (0% Phosphate).

**Figure 5 antibiotics-10-00427-f005:**
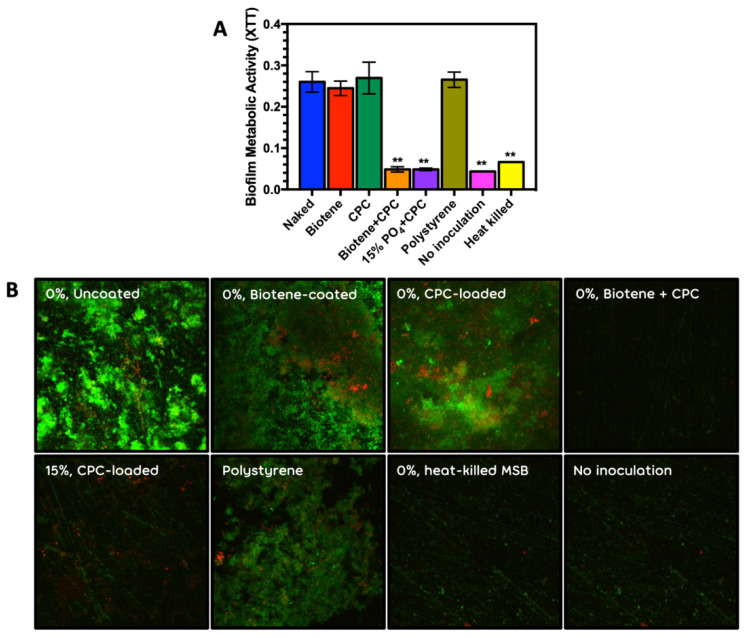
(**A**) Metabolic activity (XTT) average and standard deviations derived from 6 replicate disks and (**B**) Confocal microscopic images of mixed salivary bacterial biofilms on PMMA; 0%, Uncoated, 0%, Biotene PBF-coated, 0%, CPC-loaded, 0%, Biotene PBF + CPC, 15%, CPC-loaded, Polystyrene, 0%, heat-killed MSB and No inoculation. CPC at 350 µg/mL. ** Statistically significant reduction in metabolic activity compared to PMMA control (0% Phosphate).

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
