# Peer review of "Inhibition of Candida albicans and Mixed Salivary Bacterial Biofilms on Antimicrobial Loaded Phosphated Poly(methyl methacrylate)"

_antibiotics, 2021, doi:10.3390/antibiotics10040427_

Round 1
Reviewer 1 Report
In this manuscript the authors report the inhibition of microbial adhesion to Phosphated Poly(methyl methacrylate) in the presence or absence of antimicrobials such as cetylpyridinium chloride, KSL-W, histatin 5.
The topic is interesting. However, the authors introduce too many variables and do not deal in the right way any aspect. From the experimental point of view it is very weak. In addition, structured in this way the manuscript is confusing.
Specifically, the main issues are:
Candida strains are all clinical samples identified exclusively with the germ tube test. A more specific identification method is required, and at least one of reference strains must be added.
The concentrations used of all antimicrobials that among other things I have not found, should be five to identify if there is a dose-response relationship. Activities on individual strains should be reported.
What are the bacteria present in mixed salivary bacteria? there is no information.
Author Response
Response to Reviewer 1:
1) The topic is interesting. However, the authors introduce too many variables and do not deal in the right way any aspect. From the experimental point of view it is very weak. In addition, structured in this way the manuscript is confusing.
Response: We appreciate the reviewers remarks and suggestions for improvements of this manuscript. We gently disagree that we have introduced too many variables and do not deal in the “right way” in any aspect and are surprised by this suggestion since no justification is given for this comment. However, we have rewritten certain portions of the manuscript (using track changes) and added references to clarify our methods, results and conclusions.   Edits to the methods, results and discussion sections should now clarify that we dealt with one variable at a time in our experiments such as anionic charge (0-15% phosphate) on the naked denture base surface and demonstrated that charge alone inhibits fungal biofilm development as well as fresh mixed salivary bacteria biofilm development in a standard 24-hour in vitro biofilm model (see also Chandra 2005). Next, we assessed the effects of cationic antimicrobial peptides and CPC loaded on the charged and uncharged surfaces.  We now point out (line 177) that we used peptide concentrations that had previously been shown to inhibit candidal adhesion on various surfaces, and these references have been added to clarify why we did not pursue dose response curves for the less effective peptide antimicrobials (Moffa 2015 / Theberge 2013).   Finally, we added a salivary protein coat and an artificial saliva to assess the generalizability of the antimicrobial activity of CPC on charged and uncharged PMMA surfaces. 
2) Candida strains are all clinical samples identified exclusively with the germ tube test. A more specific identification method is required, and at least one of reference strains must be added.
Response: Thank you for sharing your opinion on that matter. Clinical isolates, with two being specifically isolated from denture stomatitis patients, was in our believe more than adequate, and the mixed salivary bacteria biofilm further strengthened the observations we made.  Furthermore, in fig 4 b one can see clear phenotypic switching of the AD-1 isolate when inoculated on a PMMA surface previously exposed to CPC.  We do not see the usefulness of running experiments on a laboratory reference strain for the clinical background of this study.
3) The concentrations used of all antimicrobials that among other things I have not found, should be five to identify if there is a dose-response relationship. Activities on individual strains should be reported.
Response: Thank you for making us aware that we have not yet included this valuable information. The concentrations run on the two peptide antimicrobials were chosen from papers that had already run dose response curves.  This was NOT made clear in the paper, and we have rectified that mistake on page 7 of the manuscript and added appropriate references.  We chose the high end of those concentrations (100 Micrograms/ml) to see if we would observe significant inhibition in the Chandra model as the Moffa and Theberge studies had previously shown in their model systems looking at C. albicans adhesion.  The CPC was run initially at this same concentration simply as a comparator, because it is an inexpensive and widely used antiseptic in over-the-counter oral rinses, and because CPC is listed as generally recognized as safe compound by the FDA. Concentrations of CPC in current mouth rinses run from 0-350 yg/ml with Crest Pro-health having the highest concentration. Our observation that fungal as well as bacterial adhesion and biofilm development in the Chandra 24-hour biofilm model was so striking that we felt that it was more useful to pursue further experiments on CPC rather than pursuing Histatin 5 and KSL-W which would be essentially repeating experiments others have done but doing so on a novel phosphate PMMA surface.  The cost of creating such a therapeutic peptide would not prove commercially viable in our opinion so we did not pursue this further, although from an academic standpoint the reviewer certainly makes a valid point.  
We believe these data are well worth publishing because they provide information that would allow a new approach to denture hygiene that can be achieved immediately and inexpensively using over the counter materials.  These observations could have significant impact on reducing aspiration pneumonia in the institutionalized elderly population where methods of denture hygiene desperately need to be improved.
4) What are the bacteria present in mixed salivary bacteria? there is no information.
Response: The bacteria present in the mixed salivary bacteria model used by Kai Leung's group include strep species, actinomyces and many other oral flora, including candida species.  We added reference [35] that details the saliva collection, clarification, bacterial isolation and resuspension to make it clear how these mixed salivary bacteria are obtained.
We thank the reviewer for helping us to improve this paper.
 
Reviewer 2 Report
This is a very significant study in a very important matter: biofilm formation on biomaterials.
Biofilms play a crucial role in the development of Candida-associated denture stomatitis, therefore inhibition of microbial adhesion to PMMA and phosphate containing PMMA has been examined in this work.
The goals of this study were to assess the effects of phosphate addition to PMMA on C. albicans and mixed salivary bacterial biofilms and to further examine biofilm metabolism on PMMA surfaces after loading one of three antimicrobials (histatin 5, KSL-W 64 and cetylpyridinium chloride) on the naked as well as salivary pre-conditioned PMMA surfaces prior to microbial adherence.
The authors concluded that the new phosphate containing EGMP-PMMA co-polymers can alter C. albicans adhesion and/or biofilm metabolism. Pre-conditioning a negatively charged phosphated PMMA surface with CPC provides a strong anti-candidal shield whereas uncharged PMMA surface is not able to benefit from CPC exposure. A salivary pellicle coating provides the necessary environment to establish an effective anti-candidal shield while not altering the phosphated PMMA from efficacious delivery of CPC.
Notwithstanding the importance of the study some minor changes should be done:
- In line 68 it is written “the Candida biofilms” and in line 77 “in Candida biofilm”. Please put Candida in italic throughout the text.
- In line 114 “C. albicans, C. glabrata” should be in italic “ albicans, C. glabrata”. Please revise all species nomenclature (they should be in italic).
- Please explain the paragraph between line 128 and line 132. I really do not understand it. It is based on which results?
- Please explain why the paragraph between line 169 and line 177 is in discussion. I think it should be rewritten.
- In line 191-192 it is written “Four clinical isolates of albicans (A1, A3, AD1, 1.1) were obtained from various sources and identified through germ tube formation”. I can only see results with AD1 and 1.1. This should be changed.
This is a very elegant study that, with minor changes, deserve to be published.
Author Response
Response to Reviewer 2:
Thank you for the kind assessment. We appreciate your time and expertise and are thankful for your positive review. We appreciate the very helpful and detailed suggestions to improve our manuscript.
1) Notwithstanding the importance of the study some minor changes should be done:
- In line 68 it is written “the Candida biofilms” and in line 77 “in Candida biofilm”. Please put Candida in italic throughout the text.
- In line 114 “C. albicans, C. glabrata” should be in italic “ albicans, C. glabrata”. Please revise all species nomenclature (they should be in italic).
Response: As requested by the reviewer we have gone through the manuscript and italicized all references to Candida as a response to their first two suggestions.
2) Please explain the paragraph between line 128 and line 132. I really do not understand it. It is based on which results?
Response: Thank you for pointing out that this section needs clarification. We appreciate the chance to improve that because it appeared, we were not clear enough in our first attempt. During the experiment we observed that with repeated use of the disks with phosphate they began to lose their inhibitory ability and we found it was due to mineral accretion. Therefore, we had to develop a better cleaning process for the disks to remove Calcium deposits. This was a phenomenon that Cao et al JDR 2010, 89(12) 1517-1521, has experienced with a carboxylated PMMA and they solved it in a similar way (Reference 32 of the manuscript).
3) Please explain why the paragraph between line 169 and line 177 is in discussion. I think it should be rewritten.
Response: The question regarding pellicle effects (lines 169-177) has been re-written, and the extraneous mentioning of the live dead staining has been removed and placed in the methods section as it should have been to begin with.
4) In line 191-192 it is written “Four clinical isolates of albicans (A1, A3, AD1, 1.1) were obtained from various sources and identified through germ tube formation”. I can only see results with AD1 and 1.1. This should be changed.
Response: Thank you for that good catch. Indeed, we are now clarifying that matter in lines 93 and 94.
Reviewer 3 Report
General Impression
The authors present a thorough description of experiments aimed to explore the physical effects of embedding negative charges on the adherence of microbes and antibiotics to the surface of PMMA resin. The experiments are well-designed, adequately described and appropriately interpreted. The importance of the findings for the development of biofilm-resistant dentures and antimicrobial surface treatments make this manuscript worthy of publication.
Specific Points
Figure 1A: The Y axis label should read XTT, not XXT.
Figure 1B: The results section describes that an increase in phosphated PMMA resin to 15% reduces surface adhesion of Candida albicans, which makes sense in combination with the reported decrease in biofilm metabolic activity (Fig 1A) and the hydrophobic nature of C. albicans biofilms. It is not clear, however, how to interpret the increase in biofilm from 15% to 25% phosphate in micrograph on the far right of Figure 1B. Is it just this reviewer's impression that the 25% phosphate photo shows higher yeast density than the 15% sample? How can it be explained?
Author Response
Response to Reviewer 3:
1) The importance of the findings for the development of biofilm-resistant dentures and antimicrobial surface treatments make this manuscript worthy of publication.
Response: Thank you for this very kind assessment. We appreciate this evaluation.
2) Figure 1A: The Y axis label should read XTT, not XXT.
Response: Very good catch. We changed accordingly.
3) Figure 1B: The results section describes that an increase in phosphated PMMA resin to 15% reduces surface adhesion of Candida albicans, which makes sense in combination with the reported decrease in biofilm metabolic activity (Fig 1A) and the hydrophobic nature of C. albicans biofilms. It is not clear, however, how to interpret the increase in biofilm from 15% to 25% phosphate in micrograph on the far right of Figure 1B. Is it just this reviewer's impression that the 25% phosphate photo shows higher yeast density than the 15% sample? How can it be explained?
Response: The second concern is regarding figure 1B and the lack of additional reduction in biofilm from the 15 to the 25% disks. This is a valid observation, and what we can say is that in this confocal sample we may have simply had a slightly higher inoculum, but we choose this figure because it came from one of our replications that did both XTT and CSLM on the same experiment and we felt we should keep the data together. It was a clear phenomenon that at 15% phosphate the disks anti-adhesion capabilities seemed to plateau. We have added supplemental figures that we think can show this better, but unless the editor tells us to change figure I B we would rather leave it as is. It does show that the charge dependent effect is NOT a clear dose response. We suspect that it may have to do with incubations in 50% saliva buffer where cation (Calcium?) deposition was something we had to address over time, and we had already added in the first revision a citation on this phenomenon (Cao et al. 2010).
3
We have also added much more detail on how we solved this problem of regenerating a truly clean surface on these disks, which we were glad to add as it will ensure that these results are repeatable. Thank you for your sharp eye!
We very much appreciate that this reviewer adds to the majority of reviewers who see value in publishing this work.
Reviewer 4 Report
Confocal images should be revised to avoid showing background, especially Fig 3F-H, Fig 4B (biothene) and Fig 5B.
Biothene alone has shown impressive anti-biofilm activity against Candida albicans in vitro. Authors should include this control to confirm the impact of the association X use of biothene alone.
Discussion ends abruptly!
There is a vast literature showing many other PMMA modifications and its impact on Candida biofilm, authors should at least mention that, and a couple of other approaches that have shown interesting results.
Authors used "Candida samples from various sources". Since those are clinical samples, please include the IRB or simmilar institutional approval number from the ethics commitee.
Please check the reference list, a few of them are not in the correct formating.
Author Response
Response to Reviewer 4:
1) Confocal images should be revised to avoid showing background, especially Fig 3F-H, Fig 4B (biothene) and Fig 5B.
Response: We appreciate this critique of our confocal pictures but would rather not make any alterations to the pictures themselves. We added more figures in a “supplemental data” file that reinforces the findings of our experiments.
2) Biothene alone has shown impressive anti-biofilm activity against Candida albicans in vitro. Authors should include this control to confirm the impact of the association X use of biothene alone.
Response: We added the complete series of Biotene experiments in a new submitted data file as “supplemental data”. Regarding the experimental design, we agree that running biotene as a separate control is crucial, and we did that, so we have added a supplement that shows all 4 candida strains, and how they are differentially effected by the Biotene pellicle, by CPC alone and the two components together. This may have been an oversight on our part, simply from space concern regarding how many figures can be in the manuscript. We think the additional CSLM figures will also address the concerns regarding background in several of the figures which showed the greatest inhibition.
3) There is a vast literature showing many other PMMA modifications and its impact on Candida biofilm, authors should at least mention that, and a couple of other approaches that have shown interesting results.
Response: We agree with these suggestions as well and have added more depth to the introduction including more background on the extensive body of work done to date on novel methods of anti-fungal surface treatments and this also spills over into the discussion which naturally addresses the abrupt end to the discussion, and we thank the reviewer for pointing this out. It clearly enhances the manuscript.
4) Discussion ends abruptly!
Response: The discussion section was revised as a result of the first rounds of reviews. We reviewed the discussion section again and made appropriate changes.
5) Authors used "Candida samples from various sources". Since those are clinical samples, please include the IRB or simmilar institutional approval number from the ethics commitee.
Response: The IRB number has been added.
4
6) Please check the reference list, a few of them are not in the correct formating.
Response: The reference list was reviewed and formatting problems with some references have been corrected. We also added numerous new references to improve the discussion section as suggested above.
We thank the reviewer for helping us to improve this paper.
Respectfully,
Round 2
Reviewer 1 Report
I appreciated the authors' response and share the importance of finding new antimicrobial strategies. Unfortunately, there are some points that cannot be overlooked.
The manuscript is certainly interesting but in my opinion it is necessary that the experiments be conducted on microorganisms deposited or identified with molecular techniques, to allow other colleagues to be able to repeat the experiments and obtain the same results.
Is there also Candida in mixed salivary bacteria ? then it is not possible to call it bacteria.
Author Response
Response to Reviewer 1:
We thank the reviewer for his willingness to review our changes based on his initial critique. The reviewer still feels after the first set of changes that we still can improve the introduction and the results presentation section, and we have made efforts to do this and have provided some supplemental materials that we think help make our case for this manuscript. The reviewer still feels that the experimental design, the methods section and the conclusions must be improved before the manuscript is ready for publication. We will address these issues below and reference our supplemental materials as well as the text alterations in the manuscript which we have altered since the last review Changes highlighted in green). We hope we have sufficiently addressed the concerns and we very much thank the reviewer for helping us bring greater clarity to our methods, and conclusions as well as expanding the background and discussion.
The reviewer has requested that we identify the microbial content of the mixed salivary bacteria model. To improve the manuscript in this area we now cite 3 additional studies that describe the major salivary sources for plaque formation as defined by Helmerhorst in 1999 [1] and more recently with improved techniques the description of the salivary microbiome, and those major species generally considered important in saliva from healthy subjects. See recent papers by Yameshita’s group in Japan (Hisayama study) [2], and a nice review by this same group [3] (see table 1 of the review). Recent findings are consistent with Helmerhorst’s model. We feel that the mixed salivary bacteria model has been a useful model for what we set out to accomplished in this work. Our aim was to evaluate anti-adhesion and subsequent biofilm development using a system that is readily accessible, well known, cited and used. This system, however, is also dynamic which addresses general applicability of our findings. The dynamism of the system reduces the usefulness of doing additional identification of the microflora as a part of this work. Since each day we gathered saliva from 3-4 healthy donors we likely had shifting microbial flora from each experiment. Trying to identify each sample seems excessive for our purposes, so we respectfully disagree that going back to identify what is in saliva is necessary. We would be repeating what others have done. We do however, feel strongly that the next series of experiments should begin to use these disks in-situ and identify what has bound to our surfaces, and if the early colonizers are altered in any way from what one would expect to bind on PMMA. That question will be central in the next series of studies and requires more funding!
The reviewer has asked for improvements in the methods section, and we have provided more detail specifically on regeneration of the disk surfaces because we felt that was important.
The reviewer said we must improve our conclusions and we have gotten more specific with some simple but important changes.
References:
- Helmerhorst EJ, Hodgson R, van 't Hof W, Veerman EC, Allison C, Nieuw Amerongen AV. The effects of histatin-derived basic antimicrobial peptides on oral biofilms. J Dent Res. 1999 Jun;78(6):1245-50
- Takeshita, T., Kageyama, S., Furuta, M., Tsuboi, H., Takeuchi, K., Shibata, Y., Shimazaki, Y., Akifusa, S., Ninomiya, T., Kiyohara, Y., Yamashita, Y., 2016. Bacterial diversity in saliva and oral health-related conditions: the Hisayama Study. Scientific Reports 6, 22164
- Yamashita, Y., Takeshita, T., 2017. The oral microbiome and human health. Journal of Oral Science 59, 201–206
Round 3
Reviewer 1 Report
The author have improve the manuscript, even if they did not fulfill all requests.Author Response
Thank you for your review.